# Physicians' perspectives on continuity of care for patients involved in the criminal justice system: A qualitative study

Latasha Jennings[1]*, Carolina Fernández Branson[2], Andrea M. Maxwell[3], Tyler N. A. Winkelman[1,4], Rebecca J. Shlafer[5]

**1** Health, Homelessness, and Criminal Justice Lab, Hennepin Healthcare Research Institute, Minneapolis, MN, United States of America, **2** School of Communication, Writing and the Arts, Department of Professional Communication, Metropolitan State University, Saint Paul, MN, United States of America, **3** Medical Scientist Training Program, Medical School, University of Minnesota, Minneapolis, MN, United States of America, **4** General Internal Medicine, Department of Medicine, University of Minnesota and Hennepin Healthcare, Minneapolis, MN, United States of America, **5** Department of Pediatrics, Medical School, University of Minnesota, Minneapolis, MN, United States of America

* latashajennings93@gmail.com

**Data Availability Statement:** All relevant data are within the manuscript and its Supporting Information files.

## Abstract

### Background

In 2016, over 11 million individuals were admitted to prisons and jails in the United States. Because the majority of these individuals will return to the community, addressing their health needs requires coordination between community and correctional health care providers. However, few systems exist to facilitate this process and little is known about how physicians perceive and manage these transitions.

### Objective

The goal of this study was to characterize physicians' views on transitions both into and out of incarceration and describe how knowledge of a patient's criminal justice involvement impacts patient care plans.

### Methods

Semi-structured interviews were conducted between October 2018 and May 2019 with physicians from three community clinics in Hennepin County, Minnesota. Team members used a hybrid approach of deductive and inductive coding, in which a priori codes were defined based on the interview guide while also allowing for data-driven codes to emerge.

### Results

Four themes emerged related to physicians' perceptions on continuity of care for patients with criminal justice involvement. Physicians identified disruptions in patient-physician relationships, barriers to accessing prescription medications, disruptions in insurance coverage, and problems with sharing medical records, as factors contributing to discontinuity of care

**Funding:** TW receives funding through a career development award from Hennepin Healthcare (https://www.hennepinhealthcare.org). LJ, TW, and RS receive financial support from the University of Minnesota Grand Challenges Research Initiative (https://strategic-planning.umn.edu/gcr-phase-2-awards). AM is funded by NIH T32 GM008244 (https://projectreporter.nih.gov/project_info_details.cfm?aid=9852690&icde=51017452). The funders had no role in study design, data collection and analysis, decision to publish, or preparation of the manuscript.

**Competing interests:** The authors have declared that no competing interests exist.

for patients entering and exiting incarceration. These factors impacted patients differently depending on the direction of the transition.

## Conclusions

Our findings identified four disruptions to continuity of care that physicians viewed as key barriers to successful transitions into and out of incarceration. These disruptions are unlikely to be effectively addressed at the provider level and will require system-level changes, which Medicaid and managed care organizations could play a leading role in developing.

## Introduction

In 2016, over 600,000 individuals were admitted to a state or federal prison in the United States (US) [1], and 10.6 million people were admitted to a US jail [2]. Nearly all of these individuals will return to the community following their incarceration [2, 3]. Previous research has shown that people in prison and jail have a high burden of chronic physical health conditions, infectious diseases, disability, mental illness, and substance use [4–9], which require ongoing health care during incarceration and upon release. People in prison or jail have higher odds of hypertension, asthma, arthritis, cervical cancer, and hepatitis compared to the general population [4]. In addition, it is estimated that 32% of individuals in prison and 40% of individuals in jail have at least one disability related to hearing, vision, cognition, mobility, self-care, or independent living [5]. Further, over half of all individuals in prison or jail have a mental health problem or meet criteria for drug dependence or abuse [6, 8]. In Hennepin County, MN, where this study was conducted, 52% of the jail population reported having a mental health problem, and 20% reported using or abusing opioids [10]. Addressing the health needs of individuals cycling in and out of prisons and jails requires coordination between community and correctional health care providers. However, this coordination is often inconsistent or nonexistent and complicated by barriers like sharing medical records between correctional and community health providers [11–13].

While some specialized programs have been developed to meet the needs of individuals leaving prison or jail, most people do not interact with these programs because they serve limited geographical areas [14–18]. The majority of individuals leaving prison or jail establish care with a community-based physician not affiliated with a re-entry program. Yet, little is known about community-based providers' understanding of transitions related to incarceration or providers' experiences with navigating care transitions for individuals who were recently incarcerated. Such information could identify opportunities to improve continuity of care for patients transitioning between the community and prison or jail.

The goal of this qualitative study was to address this gap in the literature by describing physicians' views on, and experiences with, patient transitions into and out of incarceration and how these transitions impact care plans for patients.

## Methods

Semi-structured interviews were conducted with physicians from three safety-net clinics in Hennepin County, Minnesota between October 2018 and May 2019. Hennepin County is Minnesota's largest county with a population of 1.3 million people and contains the city of Minneapolis and several large, suburban cities. These clinics were selected because they serve a

high number of patients who have histories of justice system involvement. Further, each clinic serves as a medical training institution that may be ideal sites for implementation of future physician training curriculum. This study was approved by the Institutional Review Boards at Hennepin Healthcare Research Institute and the University of Minnesota. Written informed consent was obtained from all participants.

Interview questions were developed by research team members LJ, TW, and RS, and were initially generated based on known gaps in physician training and education related to working with patients with justice system involvement. Given this gap, interview questions were created to assess physicians' knowledge of the US criminal justice system and how it impacts their patients. These questions were reviewed and refined using an iterative process by both the research team and external stakeholders with backgrounds in public health and criminal justice. The interview questions were further refined after conducting two pilot interviews. The final interview guide covered five main topics: 1) knowledge of the criminal justice system; 2) medical education and training related to the criminal justice system and working with patients involved in the criminal justice system; 3) if and how physicians screen patients for criminal justice system involvement; 4) the physical, mental, and substance-related health needs of patients with criminal justice system involvement; and 5) recommendations on how the health care system should meet the needs of patients with histories of criminal justice involvement.

After receiving approval from the medical director at each institution from which we recruited, we extended invitations to physicians by email to participate in the study. We used purposive sampling to recruit physicians within the specialties of internal medicine, psychiatry, family medicine, and emergency medicine. In recruitment materials, participants were told that the purpose of the study was to better understand physicians' knowledge and perceptions of the criminal justice system and its impact on patients to inform development of training curricula and programs that better meet the needs of providers and patients.

A total of 47 physicians were directly contacted by LJ. Of those 47 physicians, 27 (57%) responded to our inquiry, and nearly all of those who responded to our inquiry agreed to participate in a one-time interview ($n = 23$, 85%). Participants identified as predominantly White ($n = 19$, 83%), non-Hispanic ($n = 21$, 91%) and male ($n = 16$, 70%), with an average age of 48 years (range = 36–75).

All interviews were facilitated by LJ, a female project coordinator with a Master's degree in Public Health and graduate-level experience and training in conducting qualitative research. A student notetaker was also present during interviews. Interviews lasted between 36 and 90 minutes, and no repeat interviews were conducted. Interviews were conducted in private offices at participants' workplaces. One interview was conducted at a public coffee shop at the participant's request. Following each interview, LJ compiled field notes to record observations. Field notes were used to record information regarding interview setting and length along with notes on interviewee responses and body language. These notes were used to support transcription of interview audio recordings. Data collection concluded upon agreement amongst the research team that no new findings were emerging and data saturation had been reached. Interviews were audio recorded and transcribed verbatim using a web-based transcription service. All transcripts were reviewed for accuracy by members of the study team. Transcripts were not returned to participants for comment or correction.

A thematic analysis approach was utilized to identify patterns within the data [19]. For this thematic analysis, both deductive and inductive coding was used [19, 20]. Codes are words or short phrases that are used to summarize key features of the data [19]. Deductive coding was conducted by applying a priori codes that were defined based on the interview guide. For example, "knowledge of the criminal justice system" was one a priori code used to organize

data related to physicians' ability to define criminal justice terminology (i.e., prison, jail, probation parole). Inductive coding was conducted by allowing data-driven codes that were not predefined to emerge [19, 20]. "Continuity of care," the focus of this manuscript, was one such inductive code. The codes were organized into a codebook from which we identified patterns, or themes, that were present across the data [21].

Three transcripts were independently coded by LJ, RS, and CFB and a preliminary codebook containing both deductive and inductive codes was developed [21]. This preliminary codebook was independently applied to one transcript by LJ, RS, and CFB, and the team met to discuss codebook revisions. An iterative process was used to further refine the codebook, in which a subset of transcripts were independently coded by two of the coders and compared until the researchers were confident that a consensus on the meaning of all codes had been reached. The codebook was independently applied by LJ, RS, and CFB to the remaining transcripts. The codebook was ultimately refined to 18 codes (12 deductive and 6 inductive). From these codes, multiple themes were identified, and just one code, "continuity of care," is the focus of this paper. LJ, RS, and CFB identified, "continuity of care," as warranting further sub-analysis given the complexity of the data contained within this code. This sub-analysis revealed distinct patterns related to continuity of care, which was not one of the guiding questions of this research project. Those themes are described herein. NVivo (version 12) was used for all coding and analyses. Participants did not provide feedback on the findings.

## Results

Four themes emerged from our interviews with physicians related to care transitions and factors that physicians perceived as contributing to discontinuity of care for patients entering and exiting incarceration: (1) disruptions in patient-physician relationships, (2) barriers to accessing prescription medications, (3) disruptions in insurance coverage, and (4) problems with sharing medical records. Physicians discussed how these factors impacted patients differently depending on whether the patient was transitioning from the community to prison or jail, or transitioning from prison or jail back to the community. In some instances, physicians' knowledge of a patient's past or upcoming incarceration played a significant role in informing the physician's care plan for the patient.

### Disruptions in patient-physician relationships

Physicians noted changes in medical providers as an issue impacting patients entering and leaving incarceration. In most cases, physicians in the correctional setting were not the same physicians patients saw in the community setting. Changes in medical providers introduced barriers to care coordination due to limited communication between the two settings. Participants emphasized the need for transitional care planning for patients upon their release to improve communication between correctional and community health care providers. For example, one physician remarked:

> "For patients who are currently incarcerated and are going to be changing their living situation, I would love it if we could do a transitional care visit with their primary care provider." (Participant 21)

There were rare exceptions in which a community-based provider also provided correctional health care. Two physicians shared how their roles in both correctional and community settings provided unique opportunities to maintain the patient-provider relationship. As expressed by one of these physicians:

"There would be people that. . .I'd seen 'em in the jail. And then they would just come over [to community clinic], and I'd keep 'em . . . So we had continuity of care." (Participant 8)

Physicians with roles in both correctional and community settings were able to continue their care plans for patients during transitions between correctional and community settings, while physicians without access to correctional settings experienced gaps in patients' treatment plans. This resulted in negative health impacts for patients, especially those experiencing chronic physical and mental health conditions that require long-term care.

### Barriers to accessing prescription medications

Physicians reflected on patients' challenges to continue their prescription medications around periods of incarceration. Physicians expressed that their patients often experienced barriers to accessing medications for common physical chronic conditions, as well as unique barriers related to medications for opioid use disorder treatment.

Physicians shared instances in which medications that had been prescribed to their patients in the community were confiscated or otherwise discontinued when the patient became incarcerated. For instance, one physician said:

"I've had a number of patients tell me that their medications were taken away when they were arrested or brought into custody . . ." (Participant 19)

Physicians prescribing medications for opioid use disorder treatment, like buprenorphine and methadone, were particularly attuned to their patients' criminal justice system involvement due to patients being unable to continue such medications during their incarceration. In some instances, physicians made intentional changes to their care plans if a patient believed they may be incarcerated in the near future. One physician shared their approach to slowly tapering a patient off their medication for opioid use disorder in an instance where the patient was facing upcoming incarceration. While not the physician's ideal approach, the option to continue this patient's medication while incarcerated was not available:

"They know that incarceration's a real option or is impending for them. In which case, then it's trying to taper them off even though it's not the appropriate medical care for them, but because they're not going to be able to get it while they're incarcerated regardless. Then, in order to make that transition easier for the patient, then either I or they will . . . request to be tapered down to a much lower [dose] or off of their medication." (Participant 6)

For patients returning to the community from a period of incarceration, physicians expressed that patients often faced barriers to getting their medications after release. Patients were generally released from prison or jail with either no medications or a limited supply. In addition, patients often experienced financial barriers to refilling their medications once they returned to the community. One physician with experience working in a correctional health care setting shared how they prescribed affordable medications to reduce the financial burden upon release:

"I learned that it is really important to, when you're treating people with medications for chronic illnesses in prison, it's your responsibility to make sure you treat them with the best medication at the lowest price. Because if you get them stable in prison, you want them to be able to continue their medical care outside of prison." (Participant 22)

## Disruptions to insurance coverage

Physicians were also attuned to disruptions related to insurance coverage for patients who had been incarcerated. Notably, patients on public health insurance experienced suspension or termination of their health insurance when they were admitted to prison or jail, and physicians expressed frustration with this practice and acknowledged the barriers this creates for patients, especially upon their release.

One physician shared their experience with a patient foregoing needed health care because they lacked health insurance following a recent incarceration:

> "The fact that MA [Minnesota's Medicaid program] gets turned off and people are getting out of custody without . . .having insurance . . . they . . . come back and they're like, 'Yeah, I've been off my meds six months because I just didn't get around to getting insurance back.'" (Participant 3)

Thus, physicians were tasked with adapting care plans to treat complex medical problems with as little financial burden on uninsured patients as possible. As shared by another physician:

> "That was a big issue for a number of people that I saw coming out of jail is that they didn't have . . .health insurance, and they came out with some pretty . . .significant health care needs . . .that we were trying to manage . . .without health insurance." (Participant 11)

Physicians perceived that having continuous access to health insurance would reduce financial barriers to accessing health care and result in improved health outcomes for patients.

## Problems with sharing medical records

Community-based physicians also had limited access to correctional health care records, which presented challenges for physicians trying to review the care that patients received while incarcerated. As stated by one physician:

> "Trying to find out what kind of evaluation and treatment and sort of what happened when they were incarcerated. [It] can be really hard to get those medical records." (Participant 11)

Physicians expressed delays in receiving medical records from correctional facilities, and if medical records were received, they often lacked adequate information on what care was delivered during incarceration. For example, one physician stated:

> "I have gotten records from [the] prison health care system previously, but not easily . . .definitely not promptly. It's always me asking and then receiving [the records] sometime later. It's not always clear from those records what happened while the patient was there." (Participant 23)

Thus, physicians were tasked with creating patient care plans with incomplete information on what care was received during a patient's incarceration.

## Discussion

Physicians identified multiple ways in which incarceration disrupts continuity of care for patients transitioning between the community and prison or jail. Barriers to maintaining

continuity of care included disruptions in patient-physician relationships during periods of incarceration, barriers to medication access (especially those for opioid use disorder), lapses in health insurance coverage, and difficulties obtaining medical records from correctional facilities. Thus, physicians were tasked with developing care plans for patients with histories of incarceration without complete information on the health care services patients did or did not receive during their incarceration. Further, physicians recognized that patients leaving prison or jail without health insurance might have difficulty affording medical visits and medications to address their complex health needs. Given these system failures identified by physicians, physicians implemented ad hoc solutions, for example, adapting care plans to reduce the financial burden of health care costs on uninsured patients.

The barriers to continuity of care identified by providers, including disruptions to patient-provider relationships, medication access and health insurance coverage, align well with findings from previous research that explored experiences of individuals recently released from incarceration, as well as physicians who work in jail or prison settings [22–25]. Such disruptions to care may contribute to poor health outcomes and an increased risk of recidivism. Previous research has found that maintaining a continuous relationship with a primary care provider is associated with a lower likelihood of return to prison or jail [26], suggesting that this relationship plays an important role in addressing health-related factors that contribute to recidivism. The current study adds the perspectives of community-based physicians to this existing body of literature, highlighting the barriers physicians face to providing care to patients recently released from prison or jail and their approaches to mitigating these barriers. In particular, this study finds, in lieu of functioning systems to support these transitions, physicians took it upon themselves to troubleshoot issues with little to no support for such complex transfers of care.

Physicians in our study were also concerned that many correctional facilities did not provide standard of care, particularly for patients with opioid use disorder. While access to medications for opioid use disorder has expanded, there are still many jails and prisons that do not provide this treatment to patients [27], and obtaining information on the particular policy a certain prison or jail has regarding treatment for opioid use disorder can be difficult. Further, physicians shared the difficulties their patients faced accessing community-based health care without health insurance, which suggests that linkage to health insurance is needed during transitions to the community as well.

## Strengths and limitations

Although this study is the first of its kind to identify perspectives of community-based physicians about the barriers to treating patients with a history of criminal justice involvement, it does have several limitations. First, only physicians from three clinics in one geographic region and from a limited number of subspecialties were included in the study, which may limit generalizability to other health care settings and medical subspecialties. Second, sampling bias may have been introduced due to physicians with a strong interest in the study topic potentially being more likely to respond to interview invitations. Third, this study lacks the inclusion of patient voices and perspectives. Thus, we do not know if patients' perspectives on barriers to continuity of care align with those of the physicians interviewed, which would have important implications for implementing patient-centered care and should guide the development of any physician training. Finally, this study focused on transitions between the community and incarceration. However, individuals involved with the criminal justice system also make transitions within the criminal system, including transfers between jails and prisons for example, that may also impact their health. Despite these limitations, this study contributes vitally

needed data to fully assess, and thus eventually mitigate, barriers to care in this specialized patient population.

## Implications

Physicians' ad hoc approaches to addressing care transitions for patients entering or leaving jail or prison, suggests that there is a lack of systems level coordination to support these transitions. Findings from this study indicate that there is a need to develop a system of care that ensures proper transitions of care between correctional facilities and community health care providers. Currently, specialized transition programs exist to facilitate transfer of care to community providers [14–18]. However, these specialized programs serve limited geographical areas and are not accessible to all individuals transitioning between prison or jail and the community.

In order to improve health outcomes and reduce recidivism, systems-level solutions are required to link care between community settings and prisons and jails. Physicians' reports of their difficulties communicating with correctional health care providers and obtaining medical records from correctional facilities suggests that increased communication channels between the two settings are needed. This could include being more transparent about what services are offered in the correctional setting so that community providers know what patients have access to during their incarceration and more readily sharing medical records when consent is obtained from patients [28]. Because patients may not yet have an established relationship with a community provider at the time of release, they should also be provided access to free copies of their medical records from correctional health care providers to help reduce barriers to establishing care in the community.

Given the barriers patients face gaining access to health insurance coverage upon release from prison or jail, correctional health care agencies should consider partnering with health agencies to build systems to support patients upon release. This could include ensuring that patients leaving prison or jail are enrolled in health insurance coverage that will cover their medical expenses when they return to the community. This could be done through partnerships with community-based organizations that provide health insurance navigation services. Enrollment in health insurance, combined with improved medical record sharing, would allow patients to quickly establish care with a new provider in the community and receive medications they may need in a timely manner.

Medicaid and managed care organizations that administer Medicaid could play a key role in developing these system-level supports. Access to Medicaid for low-income adults was expanded under the Affordable Care Act, with most individuals released from prisons likely to be eligible for such coverage [29]. Medicaid expansion provides an opportunity to link individuals who are incarcerated to health insurance coverage prior to release and coordinate care during their transition back to the community. The barriers identified by physicians in this study suggest that coordination of care for this population through a Medicaid managed care organization would alleviate barriers to medication access, lapses in health insurance coverage, and difficulties obtaining medical records from correctional facilities. Such system-level support may lead to improved health outcomes, including a reduced risk of death, as well as reduced likelihood of recidivism for individuals with a history of incarceration.

The findings from this study highlight several important areas where further research is needed. Future research should implement and test system-level supports, such as those described herein, for patient transitions between incarceration and the community to examine their impact on health and well-being for individuals with a history of incarceration. Further, because this study was conducted in one urban, Midwest county, future research should

explore rural physicians' perceptions of the impact of incarceration on continuity of care for patients. Finally, additional research is also needed to better understand the impact of transitions within the criminal justice system, such as transitions from jail to prison, and how this impacts continuity of care and overall health.

## Supporting information

**S1 File. This is the S1 interview guide.**
(DOCX)

**S2 File. This is the S2 interview transcripts.**
(ZIP)

## Author Contributions

**Conceptualization:** Tyler N. A. Winkelman, Rebecca J. Shlafer.

**Formal analysis:** Latasha Jennings, Carolina Fernández Branson, Rebecca J. Shlafer.

**Funding acquisition:** Tyler N. A. Winkelman, Rebecca J. Shlafer.

**Investigation:** Latasha Jennings.

**Methodology:** Latasha Jennings, Carolina Fernández Branson, Tyler N. A. Winkelman, Rebecca J. Shlafer.

**Project administration:** Latasha Jennings, Rebecca J. Shlafer.

**Resources:** Tyler N. A. Winkelman, Rebecca J. Shlafer.

**Supervision:** Tyler N. A. Winkelman, Rebecca J. Shlafer.

**Writing – original draft:** Latasha Jennings.

**Writing – review & editing:** Latasha Jennings, Carolina Fernández Branson, Andrea M. Maxwell, Tyler N. A. Winkelman, Rebecca J. Shlafer.

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
