## [Decision Letter · Decision Letter 0]

5 Mar 2021

PONE-D-21-03231

Physicians' perspectives on continuity of care for patients involved in the criminal justice system: A qualitative study

PLOS ONE

Dear Dr. Jennings,

Thank you for submitting your manuscript to PLOS ONE. After careful consideration, we feel that it has merit but does not fully meet PLOS ONE’s publication criteria as it currently stands. Therefore, we invite you to submit a revised version of the manuscript that addresses the points raised during the review process.

We look forward to receiving your revised manuscript.

Kind regards,

Nickolas D. Zaller

Academic Editor

PLOS ONE

Journal Requirements:

2)  We note that you have indicated that data from this study are available upon request. PLOS only allows data to be available upon request if there are legal or ethical restrictions on sharing data publicly. For information on unacceptable data access restrictions, please see http://journals.plos.org/plosone/s/data-availability#loc-unacceptable-data-access-restrictions.

3) Please describe and/or include a copy of the interview guides used as Supporting Information file.

Reviewers' comments:

Reviewer's Responses to Questions

**Comments to the Author**

1. Is the manuscript technically sound, and do the data support the conclusions?

Reviewer #1: Partly

Reviewer #2: Yes

2. Has the statistical analysis been performed appropriately and rigorously? 

Reviewer #1: No

Reviewer #2: I Don't Know

3. Have the authors made all data underlying the findings in their manuscript fully available?

Reviewer #1: No

Reviewer #2: Yes

4. Is the manuscript presented in an intelligible fashion and written in standard English?

Reviewer #1: Yes

Reviewer #2: Yes

5. Review Comments to the Author

Reviewer #1: Thank you for the opportunity to review this manuscript. Authors are correct that there is great need to better understand physician engagement with people recently released from jails and prison. Overall, the paper is well written and takes an exploratory approach to understanding the topic area. Listed below are several areas to address to strengthen the paper:

• In the methods section, authors discuss collecting field notes – what was observed and documented in these notes? Were they analyzed and integrated into the findings presented in the paper? Additional information is needed about them.

• The analysis section needs additional detail. What specific qualitative analysis was utilized? For example, it sounds like some kind of thematic approach was used but not specifically stated. It’s important to explicitly state the approach and cite the appropriate sources. Themes don’t just emerge from the data. There is a systematic process that should be followed. Authors do a nice job describing the development of the codebook but then suggest the themes emerged from the refined codebook. People do that—make decisions about the themes and what is salient and what is not. Authors may benefit from reviewing Braun & Clark who outline, step by step, the process for conducting a thematic analysis.

• Authors write “While the majority of physicians did not provide care in correctional settings…” (p.7, line 170). No descriptions were provided about this anywhere else in the text. If authors suggest a “majority” of physicians, it’s important to describe these characteristics with the specific percentage who did work in corrections settings.

• The findings could highlight more depth in meaning. For example, on page. 7, authors identify two physicians who saw people in the community and during incarceration and the quote suggests there was continuity of care in these cases. We know generally why continuity of care is important but for this study, it would be great to detail how continuity of care is beneficial. What do physicians suggest is missing or happens to people when care is disrupted?

• Part of the stated goal of the project on page 4 says the project explores how physicians develop care plans when their patients have been recently incarcerated. The article doesn’t leave readers with much information about this. Participants discussed how physicians might alter care plans when a person is going to enter a longer period of incarceration for opioid use but there is no substantial discussion around care planning after a person exits jail or prison. This is something important to add.

• P. 14, lines 336-337: How does physicians’ use of ad hoc approaches equate to the need for more medical education? The idea that this study would inform the need for specialized training or additional training in medical school was mentioned in the introduction and I don’t think the findings from this study inform training needs in any capacity. Either additional data needs to be presented or this implication should be removed.

Reviewer #2: General comments

● Obtaining physicians’ perspectives on care of the chosen population is innovative and the topic is important

● The paper could be more concise, though was generally well-written.

● Though the research design seemed appropriate for the aims, the Methods section needs more detail and organization.

Introduction

● Clear and effectively introduces subject

● Could more completely specify characteristics of patient population served by the physicians, eg add more of a profile of incarcerated individuals in the particular geographic area

Methods

● This section is lacking adequate detail and references, and the methods are even harder to follow because it is not the order in which the research occurred.

● If applicable, the authors could identify a theoretical model and cite and then how this informed the questions that were used in the interviews. If there is not theoretical model, it would be helpful to understand more about how the authors devised the questions.

● If the journal space allows, the authors could list interview questions in a table

● The authors could elaborate on how the 47 physicians were recruited

● The authors need to cite which qualitative analytic approach was used, describe it, and cite it. There very well may be more than one cite needed in the methods section to cover all the methods used if one does not cover it all

● With this in mind, the authors could elaborate on and cite the type of approach that uses a codebook and how the four themes were arrived at

○ In particular, the authors could elaborate on and cite inductive vs. deductive coding

Results

● Appreciate presenting demographics of the participants.

● The themes are relevant, though again the methods do not adequately explain how they were chosen

● The quotes are sometimes redundant and could be edited or removed; suggest removing quotes which express a concept more than once and shorten the quotes that are used

● It is unclear how many physicians endorsed each theme/experience. Often a table is used to present the numbers of quotes in each category or theme. Eg, do the themes differ in importance? Could be more meaningful if some themes appeared to be supported more than others

● It is also common to label each quote with a pseudonym or number to enhance validity by making it clear that the quotes were from multiple individuals.

Discussion

● Consider adding subheadings to provide more detailed organization, as the journal allows. Or at least follow the order and topics in the results in order to help the readers’ orientation to the data and subsequent conclusions.

● How do the data presented in this study uniquely support changes being advocated for?

● Could more directly address how the physician perspectives in this study affect care

● Again, a quantification table could help by the authors being able to describe the percent who had a particular point of view.

Limitations

● Appreciate the authors framing limitations first with the strengths. Could call this instead “Strengths and limitations.”

● This section was very well done.

Conclusions

● Consider whether this section could instead be called “Implications” as the comments are apt but seem more consistent with this labeling.

● There appears to be a pattern of physicians going above and beyond to compensate for systematic gaps in care; this would be an interesting topic to address in this section to lead into the conclusions/implications

● Does it make sense to suggest that Medicaid/managed care organizations would be able to close the gaps in care?

○ Elaborating on lack of training in medical education might be more in line with objectives and physician perspectives or in the discussion

6. PLOS authors have the option to publish the peer review history of their article (what does this mean?). If published, this will include your full peer review and any attached files.

Reviewer #1: **Yes: **Kelli E. Canada

Reviewer #2: No

---

## [Author Response · Author response to Decision Letter 0]

31 Mar 2021

Reviewer #1:

1. In the methods section, authors discuss collecting field notes – what was observed and documented in these notes? Were they analyzed and integrated into the findings presented in the paper? Additional information is needed about them.

We have added additional information on what was documented in the field notes and how they were used to support data analysis to Page 6, lines 146-148:

Field notes were used to record information regarding interview setting and length along with notes on interviewee responses and body language. These notes were used to support transcription of interview audio recordings.

2. The analysis section needs additional detail. What specific qualitative analysis was utilized? For example, it sounds like some kind of thematic approach was used but not specifically stated. It’s important to explicitly state the approach and cite the appropriate sources. Themes don’t just emerge from the data. There is a systematic process that should be followed. Authors do a nice job describing the development of the codebook but then suggest the themes emerged from the refined codebook. People do that—make decisions about the themes and what is salient and what is not. Authors may benefit from reviewing Braun & Clark who outline, step by step, the process for conducting a thematic analysis.

We have added citations regarding our hybrid inductive and deductive coding approach to Page 6, lines 154-157. We have added a citation regarding which approach uses a codebook to Page 6, line 158. Further, we added additional information on how themes were identified to Page 6, lines 164-167:

LJ, RS, and CFB identified one code, ‘continuity of care,’ as warranting further sub-analysis given the complexity of the data contained within this code. This sub-analysis revealed distinct patterns related to continuity of care, which are the themes described herein.

3. Authors write “While the majority of physicians did not provide care in correctional settings…” (p.7, line 170). No descriptions were provided about this anywhere else in the text. If authors suggest a “majority” of physicians, it’s important to describe these characteristics with the specific percentage who did work in corrections settings.

Our goal was to highlight the two physicians who did work in both community and correctional settings. We have reworded this section on Page 8, line 213 to remove language related to a “majority” of physicians in order to focus on the two physicians that worked in both community and correctional settings and who shared how this impacted continuity of care for their patients.

4. The findings could highlight more depth in meaning. For example, on page. 7, authors identify two physicians who saw people in the community and during incarceration and the quote suggests there was continuity of care in these cases. We know generally why continuity of care is important but for this study, it would be great to detail how continuity of care is beneficial. What do physicians suggest is missing or happens to people when care is disrupted?

Thanks for this thoughtful feedback. We have added additional depth in meaning to:

Page 8, lines 222-226: Physicians with roles in both correctional and community settings were able to continue their care plans for patients during transitions between correctional and community settings, while physicians without access to correctional settings experienced gaps in patients’ treatment plans. This resulted in negative health impacts for patients, especially those experiencing chronic physical and mental health conditions that require long-term care.

Page 11, lines 313-314: Physicians perceived that having continuous access to health insurance would reduce financial barriers to accessing health care and result in improved health outcomes for patients.

5. Part of the stated goal of the project on page 4 says the project explores how physicians develop care plans when their patients have been recently incarcerated. The article doesn’t leave readers with much information about this. Participants discussed how physicians might alter care plans when a person is going to enter a longer period of incarceration for opioid use but there is no substantial discussion around care planning after a person exits jail or prison. This is something important to add.

Thank you for this feedback. In revisiting the language that we used on page 4, we believe that it is more accurate to say that our goal is to highlight the “impact” of incarceration on care plans. We have edited Page 4, lines 81-83 to: 

The goal of this qualitative study was to address this gap in the literature by describing physicians’ views on, and experiences with, patient transitions into and out of incarceration and how these transitions impact care plans for patients.

6. P. 14, lines 336-337: How does physicians’ use of ad hoc approaches equate to the need for more medical education? The idea that this study would inform the need for specialized training or additional training in medical school was mentioned in the introduction and I don’t think the findings from this study inform training needs in any capacity. Either additional data needs to be presented or this implication should be removed.

Thank you for this feedback. We have removed this implication from Page 14, line 405.

Reviewer #2: 

General comments

1. Obtaining physicians’ perspectives on care of the chosen population is innovative and the topic is important

2. The paper could be more concise, though was generally well-written.

3. Though the research design seemed appropriate for the aims, the Methods section needs more detail and organization.

Thank you! We have addressed your general comments in more detail below.

Introduction

4. Clear and effectively introduces subject

Thank you!

5. Could more completely specify characteristics of patient population served by the physicians, eg add more of a profile of incarcerated individuals in the particular geographic area

We have added health statistics available for the jail population in Hennepin County, MN, where this study was conducted. Page 3, lines 64-66:

In Hennepin County, MN, where this study was conducted, 52% of the jail population reported having a mental health problem, and 20% reported using or abusing opioids.(10)

Methods

6. This section is lacking adequate detail and references, and the methods are even harder to follow because it is not the order in which the research occurred.

We have added citations regarding our hybrid inductive and deductive coding approach to Page 6, lines 154-157. Further, we added additional information on how themes were identified to Page 6, lines 164-167:

LJ, RS, and CFB identified one code, ‘continuity of care,’ as warranting further sub-analysis given the complexity of the data contained within this code. This sub-analysis revealed distinct patterns related to continuity of care, which are the themes described herein.

We have also moved the statement regarding IRB approval to the end of the paragraph 1 on Page 4, lines 92-94 to better match the order in which this occurred relative to the other procedures described in the Methods. Similarly, we have moved information regarding interview facilitation to Page 5, lines 131-133 to better match the order in which the research occurred.

7. If applicable, the authors could identify a theoretical model and cite and then how this informed the questions that were used in the interviews. If there is not theoretical model, it would be helpful to understand more about how the authors devised the questions.

We did not use a theoretical model to develop our interview questions. We have added a sentence on how interview questions were developed to Page 4, lines 98-100:

Given this gap, interview questions were created to assess physicians’ knowledge of the US criminal justice system and how it impacts their patients.

8. If the journal space allows, the authors could list interview questions in a table

Thank you for this suggestion! The interview questions will be provided as supplementary information with our resubmission.

9. The authors could elaborate on how the 47 physicians were recruited

This is addressed in our methods section Page 5, lines 117-127:

After receiving approval from the medical director at each institution from which we recruited, we extended invitations to physicians by email to participate in the study. We used purposive sampling to recruit physicians within the specialties of internal medicine, psychiatry, family medicine, and emergency medicine. In recruitment materials, participants were told that the purpose of the study was to better understand physicians’ knowledge and perceptions of the criminal justice system and its impact on patients to inform development of training curricula and programs that better meet the needs of providers and patients. 

A total of 47 physicians were directly contacted by LJ. Of those 47 physicians, 27 (57%) responded to our inquiry, and nearly all of those who responded to our inquiry agreed to participate in a one-time interview (n = 23, 85%).

10. The authors need to cite which qualitative analytic approach was used, describe it, and cite it. There very well may be more than one cite needed in the methods section to cover all the methods used if one does not cover it all

With this in mind, the authors could elaborate on and cite the type of approach that uses a codebook and how the four themes were arrived at

a. In particular, the authors could elaborate on and cite inductive vs. deductive coding

We have added citations regarding our hybrid inductive and deductive coding approach to Page 6, lines 154-157. We have added a citation regarding which approach uses a codebook to Page 6, line 158. Further, we added additional information on how themes were identified to Page 6, lines 164-167:

LJ, RS, and CFB identified one code, ‘continuity of care,’ as warranting further sub-analysis given the complexity of the data contained within this code. This sub-analysis revealed distinct patterns related to continuity of care, which are the themes described herein.

Results

11. Appreciate presenting demographics of the participants.

Thank you!

12. The themes are relevant, though again the methods do not adequately explain how they were chosen

We added additional information on how themes were identified to Page 6, lines 164-167:

LJ, RS, and CFB identified one code, ‘continuity of care,’ as warranting further sub-analysis given the complexity of the data contained within this code. This sub-analysis revealed distinct patterns related to continuity of care, which are the themes described herein.

13. The quotes are sometimes redundant and could be edited or removed; suggest removing quotes which express a concept more than once and shorten the quotes that are used

We have edited quotes to reduce redundancy and removed one quote that did not express a unique concept. These edits were made to quotes on Page 7, lines 206-208; Page 8, lines 218-220; Page 9, lines 246-247; and Page 9 lines 250-251.

14. It is unclear how many physicians endorsed each theme/experience. Often a table is used to present the numbers of quotes in each category or theme. Eg, do the themes differ in importance? Could be more meaningful if some themes appeared to be supported more than others

Thank you for this feedback. However, we have chosen not to add a table to present the numbers of quotes in each theme given that the goal of qualitative research is to understand experiences and perspectives in a non-numerical approach. This decision is informed by Johnny Saldana’s 2011 book, Fundamentals of Qualitative Research and Lario Viljoen’s 2018 article entitled, Beyond the Numbers: Why Qualitative Data Should Not Be Used to Make Quantifying Claims--Understanding the Real Value of Qualitative Research.

15. It is also common to label each quote with a pseudonym or number to enhance validity by making it clear that the quotes were from multiple individuals.

We have added Participant numbers to each quote on Page 7, line 208; Page 8, lines 219-220; Page 9, line 247; Page 9, line 263; Page 10, line 289; Page 10, line 304; Page 11, line 311; Page 11, line 323; Page 11, line 332.

Discussion

16. Consider adding subheadings to provide more detailed organization, as the journal allows. Or at least follow the order and topics in the results in order to help the readers’ orientation to the data and subsequent conclusions.

We have reordered the Discussion to follow topics in the results on Page 13, lines 367-384.

17. How do the data presented in this study uniquely support changes being advocated for?

To date, physician perspectives on how criminal justice involvement impacts patients have not been characterized. This study adds to existing literature assessing patients’ perspectives and demonstrates an alignment in viewpoints between both patients and their physicians in the barriers to continuity of care due to transitions in and out of incarceration.

18. Could more directly address how the physician perspectives in this study affect care

We agree! We currently have another manuscript in-progress which focuses on how physician perspectives on criminal justice involvement impact patient-provider care interactions.

19. Again, a quantification table could help by the authors being able to describe the percent who had a particular point of view.

Thank you for this feedback. As stated above, we have chosen not to add a table to present the numbers of quotes in each theme given that the goal of qualitative research is to understand experiences and perspectives in a non-numerical approach.

Limitations

20. Appreciate the authors framing limitations first with the strengths. Could call this instead “Strengths and limitations.”

We agree and have renamed this section “Strengths and Limitations” on Page 14, line 406.

21. This section was very well done.

Thank you!

Conclusions

22. Consider whether this section could instead be called “Implications” as the comments are apt but seem more consistent with this labeling.

We agree with your feedback and have renamed this section “Implications” on Page 15, line 521.

23. There appears to be a pattern of physicians going above and beyond to compensate for systematic gaps in care; this would be an interesting topic to address in this section to lead into the conclusions/implications

We agree and have added a sentence to address this topic in this section to Page 15, lines 522-523:

Physicians’ ad hoc approaches to addressing care transitions for patients entering or leaving jail or prison, suggests that there is a lack of systems level coordination to support these transitions.

24. Does it make sense to suggest that Medicaid/managed care organizations would be able to close the gaps in care?

a. Elaborating on lack of training in medical education might be more in line with objectives and physician perspectives or in the discussion

Thank you for this thoughtful feedback. We too grappled with this question and felt that our themes (i.e., disruptions in patient-physician relationships, barriers to accessing prescription medications, disruptions to insurance coverage, and problems with sharing medical records) were indications of systems level problems that were outside the control of an individual physician. Medicaid and managed care organizations have the ability to provide systems level care coordination for patients with a history of incarceration that would eliminate the barriers to continuity of care identified by physicians in this study. While training on criminal justice involvement in medical education is important, education alone would not solve the barriers identified by physicians in our study.

---

## [Decision Letter · Decision Letter 1]

19 Apr 2021

PONE-D-21-03231R1

Physicians' perspectives on continuity of care for patients involved in the criminal justice system: A qualitative study

PLOS ONE

Dear Dr. Jennings,

Thank you for submitting your manuscript to PLOS ONE. After careful consideration, we feel that it has merit but does not fully meet PLOS ONE’s publication criteria as it currently stands. Therefore, we invite you to submit a revised version of the manuscript that addresses the points raised during the review process.

We look forward to receiving your revised manuscript.

Kind regards,

Nickolas D. Zaller

Academic Editor

PLOS ONE

Journal Requirements:

Reviewers' comments:

Reviewer's Responses to Questions

**Comments to the Author**

1. If the authors have adequately addressed your comments raised in a previous round of review and you feel that this manuscript is now acceptable for publication, you may indicate that here to bypass the “Comments to the Author” section, enter your conflict of interest statement in the “Confidential to Editor” section, and submit your "Accept" recommendation.

Reviewer #1: All comments have been addressed

Reviewer #2: (No Response)

2. Is the manuscript technically sound, and do the data support the conclusions?

Reviewer #1: Yes

Reviewer #2: Partly

3. Has the statistical analysis been performed appropriately and rigorously? 

Reviewer #1: Yes

Reviewer #2: Yes

4. Have the authors made all data underlying the findings in their manuscript fully available?

Reviewer #1: Yes

Reviewer #2: Yes

5. Is the manuscript presented in an intelligible fashion and written in standard English?

Reviewer #1: Yes

Reviewer #2: Yes

6. Review Comments to the Author

Reviewer #1: Authors did a nice job addressing the feedback. I have no other concerns to address and think the paper is ready for publication.

Reviewer #2: (No Response)

7. PLOS authors have the option to publish the peer review history of their article (what does this mean?). If published, this will include your full peer review and any attached files.

Reviewer #1: No

Reviewer #2: **Yes: **Diane Morse

---

## [Author Response · Author response to Decision Letter 1]

30 May 2021

Reviewer #2:

METHODS

1. For validity, at what point did Continuity of Care emerge as a theme? Was it one of your guiding questions that drove this process or did it emerge in the interviews? 

Continuity of Care emerged during the interviews and was not one of the guiding questions of this research project. Language has been added to Page 7, Lines 157-159 to clarify this point:

This sub-analysis revealed distinct patterns related to continuity of care, which was not one of the guiding questions of this research project.

Further, language has been added to Page 6, Lines 143-144 to indicate that Continuity of 

Care was an inductive code that emerged during the interviews:

“Continuity of care,” the focus of this manuscript, was one such inductive code.

2. It would be helpful to clarify that among the many themes found in the interviews, you chose Continuity of Care as the theme of focus for this paper, in Methods section, and then that you subdivided this as you already described.

Thank you for this feedback. We have added additional clarification to Page 7, Lines 154-156: 

From these codes, multiple themes were identified, and just one code, “continuity of care,” is the focus of this paper.

3. The added information on recruitment is clear and informative.

Thank you!

4. To help in understanding your analysis methods, in addition to citing the references, suggest summarizing the method in 2-3 sentences.

Additional language has been added to throughout the Methods section on Page 6, Lines 136-148 to summarize the analysis methods. The added language is underlined below:

A thematic analysis approach was utilized to identify patterns within the data.(19) For this thematic analysis, both deductive and inductive coding was used. (19, 20) Codes are words or short phrases that are used to summarize key features of the data. (19) Deductive coding was conducted by applying a priori codes that were defined based on the interview guide. For example, “knowledge of the criminal justice system” was one a priori code used to organize data related to physicians’ ability to define criminal justice terminology (i.e., prison, jail, probation parole). Inductive coding was conducted by allowing data-driven codes that were not predefined to emerge.(19,20) “Continuity of care,” the focus of this manuscript, was one such inductive code. The codes were organized into a codebook from which we identified patterns, or themes, that were present across the data.(21)

Three transcripts were independently coded by LJ, RS, and CFB and a preliminary codebook containing both deductive and inductive codes was developed.(21) 

RESULTS

1. We note that the authors chose not to list the numbers of a particular subtheme which were found in the data. While the qualitative approach is not based upon numbers, it can be helpful in the interpretation of the data. We leave this to the editors to consider in importance.

Thank you. Our response that was included in our first letter is included below for the editors’ consideration:

Thank you for this feedback. However, we have chosen not to add a table to present the numbers of quotes in each theme given that the goal of qualitative research is to understand experiences and perspectives in a non-numerical approach. This decision is informed by Johnny Saldana’s 2011 book, Fundamentals of Qualitative Research and Lario Viljoen’s 2018 article entitled, Beyond the Numbers: Why Qualitative Data Should Not Be Used to Make Quantifying Claims--Understanding the Real Value of Qualitative Research.

DISCUSSION

1. Improved organization.

Thank you!

2. Generally, the authors seem to be drawing more definitive conclusions than the data warrant, though the data are compelling. The discussion should: summarize the results, contextualize results in light of prior data, indicate where future work is needed, and carefully describe implications that are rooted in the data. In that light, consider moving some portions of discussion (the last two paragraphs-- “in order to improve health outcomes and reduce recidivism…” and “Medicaid and managed care organizations…”) to the implications section.

Thank you for this feedback. We have moved these two paragraphs to the Implications section where it is better suited. Further, we removed language from the original Implications section that is now redundant and no longer necessary due to this reorganization. These changes can be found on Pages 15-16, Lines 351-383.

3. And consider less definitive statements of “should;” these statements can be moved to the implications section as well.

We have moved these statements to the Implications section and edited the language to include statements that are less definitive than “should.” These changes can be found on Page 15, Lines 355:

This could include being more transparent about what services are offered…

Page 15, Lines 363-371:

Given the barriers patients face gaining access to health insurance coverage upon release from prison or jail, correctional health care agencies should consider partnering with health agencies to build systems to support patients upon release. This could include ensuring that patients leaving prison or jail are enrolled in health insurance coverage that will cover their medical expenses when they return to the community. This could be done through partnerships with community-based organizations that provide health insurance navigation services. Enrollment in health insurance, combined with improved medical record sharing, would allow patients to quickly establish care with a new provider in the community and receive medications they may need in a timely manner.

Page 15, Line 373:

Medicaid and managed care organizations that administer Medicaid could play a key role in…

4. This is particularly important for policy change recommendations which could perhaps be described as worthy of examination and to then study outcomes.

We have added this as a potential focus of future research to Page 16, Lines 386-388:

Future research should implement and test system-level supports, such as those described herein, for patient transitions between incarceration and the community to examine their impact on health and well-being for individuals with a history of incarceration.

LIMITATIONS

1. The authors rightly and importantly point out that rural physicians are inadequately described in this area. It could be stated as a limitation in this study and/or point in the direction for future research. 

We have added this another focus of future research to Page 16, Lines 388-390:

Further, because this study was conducted in one urban, Midwest county, future research should explore rural physicians’ perceptions of the impact of incarceration on continuity of care for patients.

---

## [Decision Letter · Decision Letter 2]

30 Jun 2021

Physicians' perspectives on continuity of care for patients involved in the criminal justice system: A qualitative study

PONE-D-21-03231R2

Dear Dr. Jennings,

We are pleased inform you that your manuscript has been judged scientifically suitable for publication and will be formally accepted for publication once it meets all outstanding technical requirements.

Kind regards,

Nickolas D. Zaller

Academic Editor

PLOS ONE

Additional Editor Comments (optional):

Reviewers' comments:

Reviewer's Responses to Questions

**Comments to the Author**

1. If the authors have adequately addressed your comments raised in a previous round of review and you feel that this manuscript is now acceptable for publication, you may indicate that here to bypass the “Comments to the Author” section, enter your conflict of interest statement in the “Confidential to Editor” section, and submit your "Accept" recommendation.

Reviewer #2: All comments have been addressed

2. Is the manuscript technically sound, and do the data support the conclusions?

Reviewer #2: Yes

3. Has the statistical analysis been performed appropriately and rigorously? 

Reviewer #2: Yes

4. Have the authors made all data underlying the findings in their manuscript fully available?

Reviewer #2: Yes

5. Is the manuscript presented in an intelligible fashion and written in standard English?

Reviewer #2: Yes

6. Review Comments to the Author

Reviewer #2: This paper has been re-reviewed and the concerns have been addressed. The authors have addressed this reviewer's concerns.

7. PLOS authors have the option to publish the peer review history of their article (what does this mean?). If published, this will include your full peer review and any attached files.

Reviewer #2: No

---

## [Editor Report · Acceptance letter]

5 Jul 2021

PONE-D-21-03231R2 

Physicians’ perspectives on continuity of care for patients involved in the criminal justice system: A qualitative study 

Dear Dr. Jennings:

I'm pleased to inform you that your manuscript has been deemed suitable for publication in PLOS ONE. Congratulations! Your manuscript is now with our production department. 

Kind regards, 

on behalf of

Dr. Nickolas D. Zaller 

Academic Editor

PLOS ONE